# Two-stage training for abdominal pan-cancer segmentation in weak label

Hanwen Zhang[1,2,], Yongzhi Huang[1,2,3,], and Bingding Huang[1,*,]

[1] College of Big Data and Internet, Shenzhen Technology University, Shenzhen, 518188, China
[2] College of Applied Sciences, Shenzhen University, Shenzhen, 518060, China
[3] School of Artificial Intelligence, Beijing University of Posts and Telecommunications, Beijing, 100876, China

**Abstract.** Constructing comprehensive labeled datasets for medical image segmentation tasks is time-consuming, requiring intensive masks annotated carefully by experienced radiologists. Existing benchmark datasets provide the necessary masks to train the supervised-based segmentation models, including single-organ datasets and multiple-organ datasets. However, it is still challenging when deploying large-scale models with a union of multiple datasets due to annotation conflicts. For example, some organ or tumor annotations are missing in most cases (weak label) in the FLARE23 challenge dataset. To overcome the limitation of segmentation models in this situation, we propose a two-stage training method to train an efficient segmentation model with weak label. In the first stage, only strong labels (complete organ labels) are used to train models by the nnU-Net, while the weak labels (incomplete organ labels) are filled by generating pseudo labels using nnU-Net. Then the lightweight coarse-to-fine network is trained using the supplemented data in the second stage. Experiments on the FLARE23 challenge (MICCAI FLARE23) demonstrate that coarse-to-fine networks can reduce computational complexity and resource consumption during the inference stage while maintaining high performance, in the case of pseudo labeled supplementary data. With a speed of 12.6 seconds per case, our proposed method achieves an average DSC of 0.8920 and an average NSD of 0.9482 on the FLARE23 validation set.

**Keywords:** Weak label · Pseudo label · Two-stage training.

## 1 Introduction

Abdominal organ segmentation is a crucial step in the clinical diagnosis of abdominal diseases. Deep learning-based segmentation methods have demonstrated the ability to efficiently and accurately identify organ boundaries, sizes, and locations, aiding doctors in rapidly identifying potential lesions and disease areas[1].

---

[1] *Corresponding author: Bingding Huang(huangbingding@sztu.edu.cn).

The family of U-Net [2] architectures is the most mainstream in deep supervised learning methods for medical image segmentation tasks. Subsequently, various CNN-based segmentation networks based on the U-Net architecture emerged, such as ResU-Net [3] and U-Net++ [4]. Meanwhile, the transformer-based models are also naturally compatible with U-Net architecture, and excellent networks such as Trans U-Net [5], Swin U-Net [6], and so on have emerged for medical image segmentation. Additionally, some works focus on improving segmentation performance by using multi-view, multi-task, and multi-scale techniques, trying complex data augmentation methods, or other tricks like multi-level feature fusion and deep supervision. The most representative framework is nnU-Net [7], which is a milestone work that achieves SOTA performance using U-Net architecture with a series of heuristic rules that can deploy and train segmentation models on any dataset automatically, demonstrating the high adaptability and robustness of its framework.

However, even such a comprehensive framework, nnU-Net, cannot be used directly for annotations with different labels in multiple datasets, which is caused by the problem of annotation conflicts. Fig. 1 gives a specific example to illustrate this problem. Specifically, weak label case (2) contains only tumor, case (3) contains tumor and some organs, and (4) includes all organs without tumor. Therefore, some organs are incorrectly annotated as background, and overlapping annotation conflicts over cases. Although partly labeled data has additional annotation information and also inherits semantic information like unlabeled data, due to annotation conflicts, the performance of models trained by multiple datasets will probably not improve or even degrade compared with models using a single dataset.

To address this issue, many attempts have been made to explore multiple weak label datasets in a more efficient manner. Fang et al. proposed a new network named Pyramid Input Pyramid Output Feature Abstraction Network (PIPO-FAN) using multi-scale features to exploit weak label proportion information [8]. Enlightened by multi-branch networks and dynamic filter learning, Zhang et al. considered multiple datasets as independent tasks and designed a single shared model, a dynamic on-demand network (DoDNet), receiving task-specific signals to avoid label conflicts [9]. A similar approach is conditional nnU-Net proposed by Zhang et al. [10], which also used special signals to control segmentation models dynamically. Different from the design of segmentation architectures, some works tried to reconsider the point of loss functions to solve label conflicts. For instance, Shi et al. proposed marginal loss and exclusion loss for weak label supervised multi-organ segmentation [11]. Furthermore, Liu et al. merged weak labeled datasets using incremental learning methods, introducing a light memory module mechanism based on marginal loss and exclusion loss to further improve and stabilize the model performance with continuously incremental datasets [12]. These methods fully used weak label datasets, enabling the deployment of a comprehensive segmentation model trained by multiple datasets simultaneously.

In the FLARE23 challenge, the dataset consists of labeled, weakly labeled, and unlabeled CT image data. As shown in Fig. 2, only 222 images have complete annotations for all organs, and the remaining 1978 cases only have annotations for specific organs. To achieve higher segmentation performance than baseline supervised learning methods, fully utilizing unlabeled data and resolving annotation conflicts caused by weak label data is a key breakthrough in this competition. To this end, we attempt to merge weak labeled data with completely labeled data and propose an efficient strategy that breaks down the barriers between weak label datasets, even existing conflicts overlapping and further alleviates the problem of developing vanilla segmentation methods combining several different benchmark datasets. We also follow the trend of the FLARE competition series, and pay attention to optimizing the resource consumption and speed in the inference phase. Based on the experience of the Flare22 challenge [13] (2022-MICCAI-FLARE), using either the nnU-Net [7] adaptive framework or the EfficientSeg [14] coarse-to-fine framework combined with a semi-supervised algorithm can effectively handle unlabeled data. We will use the two networks mentioned above to design a training framework that can use weak labels to address the abovementioned challenges.

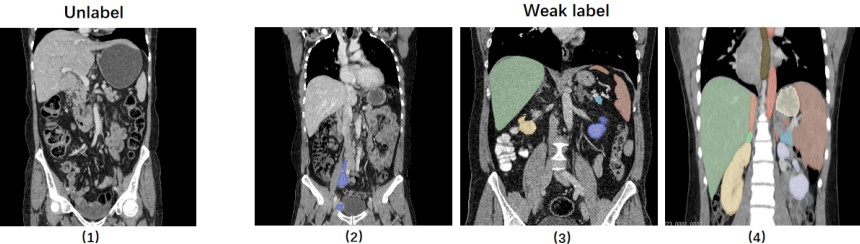

**Fig. 1.** Image(1) shows a CT without any annotation. Images (2), (3), and (4) show the weak label, where (2) has only tumor labeling, (3) contains tumor labeling and labeling of some organs, and (4) includes labeling all organs without tumors.

## 2    Method

To address the challenges posed by weak labels and imbalanced data in abdominal organ segmentation, we propose a novel training framework that utilizes statistical analysis to divide the data into different categories.

The main objective of our approach is to select relatively well-annotated strong labels from weak labels for the first round of training. We then use the model obtained from the first round of training to supplement the weak label data according to specific rules, enabling iterative training to obtain the final model.

In Section 2.2, we provide further details on our proposed approach, including the specific rules used to supplement the weak label data and the iterative training process. Our approach leverages the strengths of both the nnU-Net adaptive framework and the two-stage EfficientSeg framework, combined with semi-supervised learning algorithms, to improve the accuracy and efficiency of abdominal organ segmentation.

## 2.1   Preprocessing

Our proposed approach leverages the strengths of two networks, nnU-Net and EfficientSeg, each with its own preprocessing techniques.

nnU-Net provides a self-configuration pre-training pipeline depending on statistics information in specific datasets. To ensure the high performance of nnU-Net, we utilized this automatic preprocessing method for the FLARE23 dataset, including adjusting the target spacing and then resampling, voxel intensity normalization, and data augmentation techniques.

As for EfficientSeg, the network is a two-stage segmentation network that accepts an interpolated overall image as input, eliminating the need to adjust the image spacing. During the coarse segmentation stage, the image is interpolated and scaled to a size of [160, 160, 160]. During the fine segmentation stage, images are cropped so that only foreground regions remain and then padded to a size of [192, 192, 192] before being interpolated and scaled. The foreground information in the training process is provided by ground truths, while the one in the inference process is from masks generated from the coarse segmentation stage. The image intensity is clipped to a range of [-325, 325]. Additionally, a series of data augmentations are used in the fine segmentation stage, shown in Table 1.

**Table 1.** Data augmentation details in the fine segmentation stage.

| | |
|---|---|
| RandFlipd-x | prob=0.5 |
| RandFlipd-y | prob=0.5 |
| RandFlipd-z | prob=0.5 |
| RandZoomd | min-zoom=0.9, max-zoom=1.2, prob=0.15 |
| RandGaussianNoised | std=0.01, prob=0.15 |
| RandGaussianSmoothd | sigma=(0.5, 1.15), prob=0.15 |
| RandScaleIntensityd | factors=0.3, prob=0.15 |
| RandAdjustContrastd | prob=0.15 |

## 2.2   Proposed method

As shown in Fig. 2, statistical analysis is conducted on 2200 annotated data samples in this dataset, revealing a ubiquitous lack or omission of organ or tumor

segmentation. To address the challenges of weak labels and imbalanced data, we further analyze the distribution of annotations and propose a framework that can effectively train segmentation models with weak labels. It is worth mentioning that all unlabeled images are not used in our proposed method.

After checking category information in annotations, we found that annotations with a single category (excluding background) were mainly for the pan-cancer region segmentation. In contrast, annotations with thirteen categories mainly include regions of abdomen organs. Therefore, we split the dataset into two categories: cases with complete organ annotations (strong label) and cases with partial organ annotations (weak label).

Based on the condition of the FLARE23 dataset, our motivation is to distill knowledge from cases with strong labels, then use it to guide models to segment organs annotated wrongly as background in the weak label, and finally re-train the segmentation model with the whole annotated data. Specifically, our proposed framework consists of three stages: strong label training, weak label supplement, and retraining, as shown in Fig. 3. Each stage's network architecture is configured separately based on specific objectives and requirements. First, the strong label training stage automatically applies the self-configured framework nnU-Net to learn from the well-annotated strong label data. Second, the weak label supplement stage utilizes the EfficientSeg coarse-to-fine framework combined with semi-supervised learning algorithms to supplement the weak label data. Third, the retraining stage combines the two networks to iteratively refine the segmentation model using the supplement weak label data.

**Strong label training** In this stage, all annotated training data is split into two parts: weak label data and strong label data. Weak label data are not used due to annotation conflicts caused by missing organ annotations, resulting in degeneration and even not convergence during the training stage. To solve this problem, the strong label is selected to train nnU-Net as a teacher model that can generate credible pseudo labels for complementing annotations on missing organs. In detail, we consider 222 cases of strong label data as an independent training set and train a segmentation model through default nnU-Net 3D configuration.

**Weak label supplement** In the weak label supplement stage, we aim to utilize the nnU-Net model trained on strong label data to complement the missing annotations for organ regions. First, all cases with weak labels are inferred by nnU-Net to generate pseudo labels. We take a redundancy inference mode to obtain accurate pseudo labels, including the Test-Time Augmentation (TTA) method and connected component analysis. It is worth noting that the tumor category is not involved in the above step of pseudo-label generation. Due to poor performance and significant uncertainty in the tumor region, only 13 organ categories are predicted to complement weak labels.

Second, we replace the foreground region wrongly annotated as background in each weak label following a criterion: retaining original foreground annotations

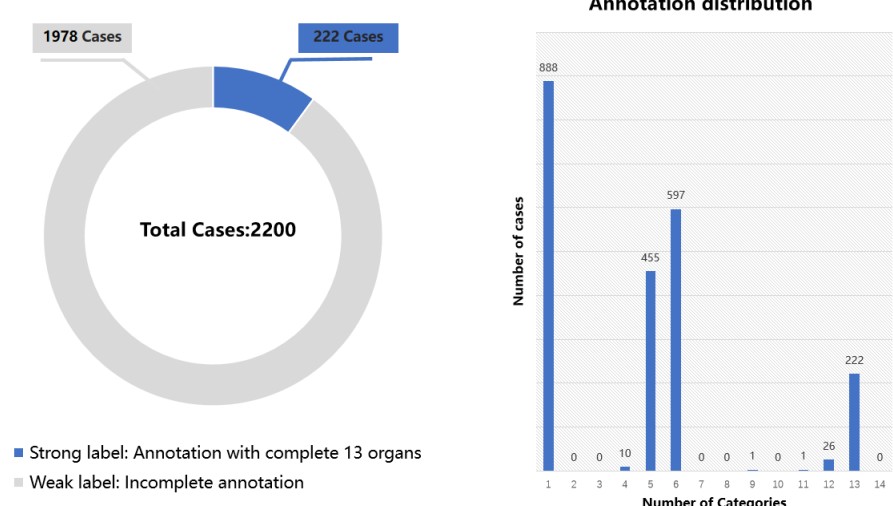

**Fig. 2.** Distribution of label counts: one important finding from our statistical analysis of the 2200 annotated data samples was that 888 of them contained only one label, which was mainly for pan-cancer region segmentation. On the other hand, the 222 samples that contained thirteen labels were primarily used for abdominal multi-organ segmentation.

in weak labels. This motivation is based on a belief that original foreground annotations have higher accuracy than predicted pseudo labels. In detail, we process each foreground category separately. The specific rules are as follows: For each foreground category in the pseudo label, if this category appears in the weak label, then the pseudo label for this category will be discarded; if the category never appears, the corresponding background region in the weak label will be replaced with this category.

**Retraining** At this stage, the two-stage EfficientSeg will be used for retraining. All annotations used in this stage are from 2200 supplement label data combined with strong label data and supplemented weak label data.

The coarse segmentation stage roughly locates the foreground region in the original image, which guides the foreground cropping for the fine segmentation stage. During the coarse segmentation training, 2200 supplement label data were used in training. Then, the fine segmentation stage further refined segmentation masks cropped from the coarse stage. During the fine segmentation training stage, we utilized supplement labels to locate the foreground as input. By utilizing the supplement labels for fine segmentation training, we achieved significantly improved segmentation accuracy and robustness in EfficientSeg.

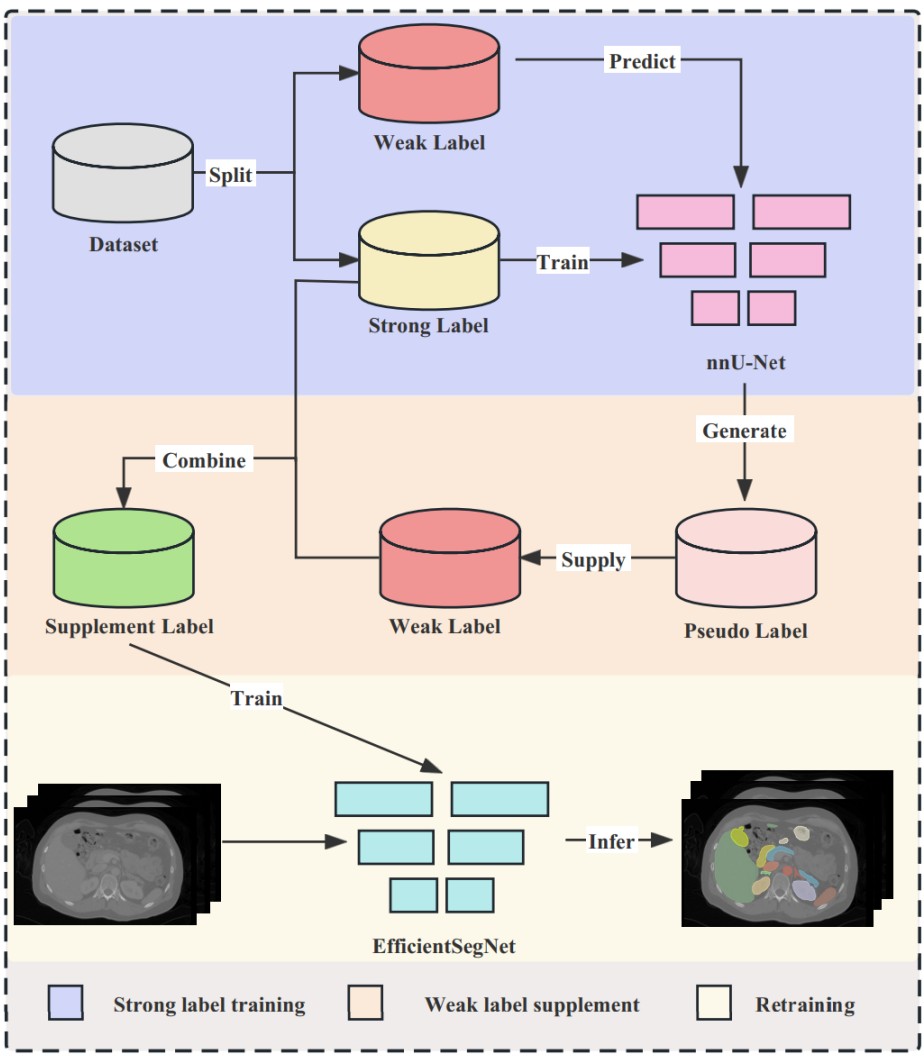

**Fig. 3.** This framework consists of three parts. Strong label training: Strong labels are selected from weak labels to be trained individually using nnU-Net. Weak label supplement: The remaining weak label is complemented using nnU-Net-generated labels. Retraining: The coarse-to-fine EfficientSegNet is trained using all the supplement labels to obtain the inference model.

**Inference speed and resources consumption trade-offs** We use a coarse-to-fine segmentation network in the inference stage to optimize the inference speed and resource usage and to avoid using a time-consuming sliding window technique. Any size image can be segmented through two inference stages by using the coarse-to-fine network. Therefore, the inference speed is improved sig-

nificantly compared with one-stage segmentation models with the sliding window technique. Following the EfficientSeg implementation, anisotropic convolution, anisotropic pooling, and FP16 are also used to reduce GPU memory usage, which is discussed in detail in [14].

### 2.3   Post-processing

We employed TTA to improve the final segmentation results during the strong label training stage. Additionally, final segmentation will adaptively keep the largest connected region to reduce false positives. Meanwhile, the coarse and fine segmentation results are also refined by the connected region analysis.

## 3   Experiments

### 3.1   Dataset and evaluation measures

The FLARE 2023 challenge is an extension of the FLARE 2021-2022 [15] [16], aiming to promote the development of foundation models in abdominal disease analysis. The segmentation targets cover 13 organs and various abdominal lesions. The training dataset is curated from more than 30 medical centers under the license permission, including TCIA [17], LiTS [18], MSD [19], KiTS [20,21], and AbdomenCT-1K [22]. The training set includes 4000 abdomen CT scans where 2200 CT scans with weak label and 1800 CT scans without label. The validation and testing sets include 100 and 400 CT scans, respectively, which cover various abdominal cancer types, such as liver cancer, kidney cancer, pancreas cancer, colon cancer, gastric cancer, and so on. The organ annotation process used ITK-SNAP [23], nnU-Net [24], and MedSAM [25].

The evaluation metrics encompass two accuracy measures—Dice Similarity Coefficient (DSC) and Normalized Surface Dice (NSD)—alongside two efficiency measures—running time and area under the GPU memory-time curve. These metrics collectively contribute to the ranking computation. Furthermore, the running time and GPU memory consumption are considered within tolerances of 15 seconds and 4 GB, respectively.

### 3.2   Implementation details

**Environment settings** The development environments and requirements are presented in Table 2.

**Dataset split** There is no multiple cross-validation for the training of nn-Unet and EfficientSegNet. For the nnU-net, 20% of 222 cases was randomly selected as the validation set. For the EfficientSegNet, 100 cases in 2200 cases were randomly selected as the validation set.

**Table 2.** Development environments and requirements.

| | |
|---|---|
| System | Ubuntu 20.04.1 LTS |
| CPU | AMD EPYC 7742 64-Core Processor |
| RAM | 1.8TB |
| GPU | 8 NVIDIA A100 (40G) |
| CUDA version | 11.7 |
| Programming language | Python 3.10 |
| Deep learning framework | torch 1.10, monai 1.0 |
| Code | https://github.com/XIANYUNYEHE-DEL/two-stage-retraining-seg |

**Training protocols** In both the strong label training stage and retraining process of our proposed framework, we utilized three different models with different configurations to improve segmentation accuracy. The protocols of these models are shown in Table 3. In the training stage of nnU-Net, relevant hyperparameters are automatically generated according to its adaptive rules. Patch size is fixed as 32 * 128 * 192 (D * W * H) and network training using SGD with a learning rate of 0.01 for 1000 epochs. As for EfficientSegNet, training will be divided into coarse model training and fine model training. In the coarse model training stage, batch size is set to 2 and patch size is fixed as 160 * 160 * 160 (W * H * D). Optimizer in the training is used AdamW with 0.01 learning rate and 0.00001 weight decay. First 50 epochs used as warm-up and using 500 epochs for the training with Cosine Annealing strategy. Loss function is selected to Dice and Cross-Entropy. In the fine model training stage, Most of the settings have not been modified. Patch size is fixed as 192 * 192 * 192 (W * H * D) and training epochs reduced to 300 for saving training time.

**Table 3.** Training and Inference protocols.

| Stage | Pseudo labeling | Coarse model | Fine model |
|---|---|---|---|
| Mode | nnU-Net 3D | 3D U-Net | EfficientSegNet |
| Network initialization | "he" normal initialization | "he" normal initialization | "he" normal initialization |
| Batch size | 2 | 2 | 2 |
| Patch size | 48×192×192 | 160×160×160 | 192×192×192 |
| Total epochs | 1000 | 500 | 300 |
| Optimizer | SGD | AdamW | AdamW |
| Weight decay | 3e-5 | 1e-5 | 1e-5 |
| Initial learning rate (lr) | 0.01 | 0.01 | 0.01 |
| Lr scheduler | ReduceLROnPlateau | Warmup and Cosine Annealing | Warmup and Cosine Annealing |
| Training time | 72 hours | 24 hours | 36 hours |
| Loss function | Dice and Cross-Entropy | Dice and Cross-Entropy | Dice and Cross-Entropy |

## 4 Results and discussion

### 4.1 Quantitative results on validation set

We used EfficientSegNet, which was trained directly using 2200 cases of labeled data as the baseline. nnU-Net, which was trained using 222 cases (strong label)

containing all organ segmentations, and EfficientSegNet, which was trained using our weak label training framework, were compared with baseline on public validation, respectively. The quantitative results are shown in Table 4.

**Table 4.** Quantitative evaluation results for ablation study on online validation.

| Target | baseline | | nnU-net(222) | | EfficientSegNet | |
|---|---|---|---|---|---|---|
| | DSC(%) | NSD(%) | DSC(%) | NSD(%) | DSC(%) | NSD(%) |
| Liver | 94.72 | 93.03 | 96.58 | 98.34 | 97.45 | 98.63 |
| Right Kidney | 87.81 | 85.23 | 93.12 | 94.29 | 93.36 | 94.11 |
| Spleen | 91.55 | 91.51 | 96.04 | 97.35 | 96.71 | 98.29 |
| Pancreas | 02.08 | 02.37 | 84.69 | 96.29 | 84.67 | 95.19 |
| Aorta | 85.36 | 84.95 | 96.28 | 98.62 | 95.85 | 98.68 |
| Inferior vena cava | 71.92 | 63.29 | 94.47 | 96.24 | 93.78 | 96.17 |
| Right adrenal gland | 02.00 | 02.00 | 83.08 | 95.39 | 81.18 | 94.64 |
| Left adrenal gland | 01.00 | 01.00 | 80.59 | 93.18 | 78.55 | 92.03 |
| Gallbladder | 10.00 | 10.00 | 82.93 | 82.01 | 85.38 | 85.87 |
| Esophagus | 00.00 | 00.00 | 83.17 | 93.47 | 82.79 | 93.53 |
| Stomach | 04.73 | 03.55 | 92.71 | 96.65 | 92.67 | 96.66 |
| Duodenum | 31.19 | 55.92 | 84.84 | 95.97 | 83.76 | 95.27 |
| Left kidney | 87.84 | 91.34 | 84.95 | 92.29 | 93.13 | 93.55 |
| Tumor | 05.48 | 01.68 | 00.00 | 00.00 | 29.98 | 20.49 |
| Average | 43.83 | 44.47 | 89.22 | 94.62 | 89.20 | 94.82 |

We observed that using weak labels for direct training often resulted in poor labeling quality, which can negatively impact the training process and lead to eventual failure. We decomposed the task into three stages to address this issue: strong label training, weak label supplement, and retraining.

For strong label training, we utilized nnU-Net, a well-established segmentation model trained on a dataset of 222 cases containing all organ segmentations with strong labels. Our experiments showed that nnU-Net achieved a Dice similarity coefficient (DSC) of 0.892, indicating that it is effective in organ segmentation. We then used the organ segmentation results obtained from nnU-Net as a generative network for organ pseudo-label.

We used EfficientSegNet to train on the 2200 cases with pseudo-label for retraining. Our experiments showed that EfficientSegNet achieved an average DSC of 0.892 for all organs and a tumor DSC of 0.299.

## 4.2 Qualitative results on validation set

Figure 4 shows the segmentation results for the baseline and our method. Among the results in case#0047 and case#0070, our method can accurately segment organs and identify tumor regions and make precise judgments even for segmentation at the boundaries of some small organs. However, in case#0029 and case#0035, our method shows some false-negative determinations of the tumor

**Table 5.** Quantitative evaluation results.

| Target | Public Validation | | Online Validation | | Testing | |
|---|---|---|---|---|---|---|
| | DSC(%) | NSD(%) | DSC(%) | NSD(%) | DSC(%) | NSD (%) |
| Liver | 97.46 ± 1.03 | 94.61 ± 5.13 | 97.45 | 98.63 | 96.38 | 96.99 |
| Right Kidney | 90.46 ± 19.95 | 87.09 ± 20.45 | 93.36 | 94.11 | 93.89 | 93.49 |
| Spleen | 95.83 ± 8.02 | 96.87 ± 7.34 | 96.71 | 98.29 | 96.02 | 97.41 |
| Pancreas | 86.09 ± 7.24 | 83.09 ± 11.89 | 84.67 | 95.19 | 88.87 | 96.62 |
| Aorta | 95.03 ± 2.98 | 95.42 ± 5.30 | 95.85 | 98.68 | 96.16 | 99.15 |
| Inferior vena cava | 92.70 ± 3.84 | 89.18 ± 6.59 | 93.78 | 96.17 | 94.32 | 97.06 |
| Right adrenal gland | 77.31 ± 20.09 | 89.80 ± 19.19 | 81.18 | 94.64 | 80.57 | 94.22 |
| Left adrenal gland | 77.35 ± 16.86 | 88.95 ± 19.38 | 78.55 | 92.03 | 79.63 | 93.15 |
| Gallbladder | 79.84 ± 28.03 | 79.85 ± 29.78 | 85.38 | 85.87 | 81.92 | 84.31 |
| Esophagus | 81.58 ± 16.65 | 83.45 ± 17.22 | 82.79 | 93.53 | 88.26 | 97.75 |
| Stomach | 93.21 ± 3.89 | 90.62 ± 10.57 | 92.67 | 96.66 | 92.18 | 96.26 |
| Duodenum | 83.49 ± 6.38 | 79.70 ± 9.00 | 83.76 | 95.27 | 86.20 | 96.24 |
| Left kidney | 91.43 ± 14.96 | 87.72 ± 17.00 | 93.13 | 93.55 | 92.96 | 93.34 |
| Tumor | 35.39 ± 34.80 | 24.97 ± 28.54 | 29.98 | 20.49 | 39.64 | 26.51 |
| Organ average | 84.08 ± 22.19 | 83.67 ± 23.83 | 89.20 | 94.82 | 89.69 | 95.02 |

region. The locations marked in the red box in the diagram show some false-negative situations. The blue area in the box is the pan-cancer area label. It can be observed that our method always wrongly classified tumor regions as normal organs. The reason may be that there is tumor regions in the supplemented organ label area, but our method has no suitable strategy to correct it.

### 4.3   Segmentation efficiency results on validation set

The efficiency test results are shown in Table 6. Using less than 4GB of GPU Memory, our method can also infer larger images in less than 20 seconds.

**Table 6.** Quantitative evaluation of segmentation efficiency in terms of the running them and GPU memory consumption. Total GPU denotes the area under the GPU Memory-Time curve. Evaluation GPU platform: NVIDIA QUADRO RTX5000 (16G).

| Case ID | Image Size | Running Time (s) | Max GPU (MB) | Total GPU (MB) |
|---|---|---|---|---|
| 0001 | (512, 512, 55) | 12.6 | 3032 | 10455 |
| 0051 | (512, 512, 100) | 9.04 | 2956 | 9044 |
| 0017 | (512, 512, 150) | 9.64 | 2970 | 10245 |
| 0019 | (512, 512, 215) | 11.39 | 2994 | 13420 |
| 0099 | (512, 512, 334) | 11.33 | 3172 | 13395 |
| 0063 | (512, 512, 448) | 14.58 | 3400 | 20043 |
| 0048 | (512, 512, 499) | 14.6 | 3254 | 19642 |
| 0029 | (512, 512, 554) | 16.87 | 3938 | 24931 |

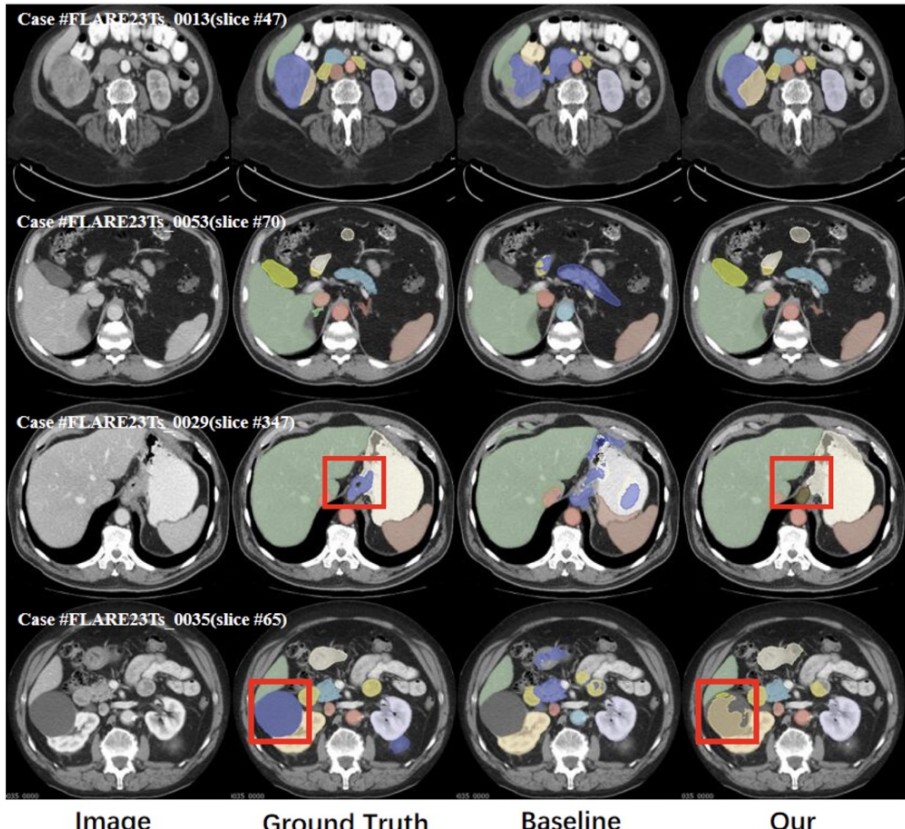

**Fig. 4.** Qualitative results of on good (#0047 and #0070) and bad (#0029 and #0035) cases. The first column is the image, the second is the ground truth, the third is the Baseline results, and the fourth is the predicted results by our method.

### 4.4    Results on final testing set

The test results are shown in Table 5. In the test dataset, we achieved an average DSC of 0.8969 and NSD of 0.9502 for all organs. This is reliable for organ segmentation. At the same time, the average inference time of our method is less than 10s with few resources. However, there are great limitations on the segmentation effect of tumors.

### 4.5    Limitation and future work

The two-stage coarse-to-fine model used in our proposed framework maintains high inference speed while achieving a high level of segmentation performance. However, we found that the performance of tumor segmentation was worse than that of abdominal organs. After an elaborate analysis of bad cases, we found that

tumors are annotated as irregular regions with non-smooth edges. In contrast, the predicted ones are probably smoothed into sphere-like regions after the resizing operation, resulting in an unnegligible error in the edge. In future work, we will further investigate segmentation models with high performance on tumors. One solution is to replace the traditional resizing operation with learning-based methods like correlation interpolation.

## 5     Conclusion

This paper proposed a two-stage training approach to overcome the problem that weak label data cannot be used for training general segmentation models directly. A pseudo-label generating network is trained using those cases with strong labels in the first training. After supplementing all weak label data using pseudo labels, the coarse-to-fine network is retrained for the inference stage. Under the limitation of computing resources, experimental results show that our method fully uses weak label data and performs well in segmentation and inference speed.

**Acknowledgements** We would like to thank the School-Enterprise Graduate Student Cooperation Fund of Shenzhen Technology University. The authors of this paper declare that the segmentation method they implemented for participation in the FLARE 2023 challenge has not used any pre-trained models nor additional datasets other than those provided by the organizers. The proposed solution is fully automatic without any manual intervention. We thank all the data owners for making the CT scans publicly available and CodaLab [26] for hosting the challenge platform.

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

**Table 7.** Checklist Table. Please fill out this checklist table in the answer column.

| Requirements | Answer |
|---|---|
| A meaningful title | Yes |
| The number of authors ($\leq 6$) | 3 |
| Author affiliations, Email, and ORCID | Yes |
| Corresponding author is marked | Yes |
| Validation scores are presented in the abstract | Yes |
| Introduction includes at least three parts: background, related work, and motivation | Yes |
| A pipeline/network figure is provided | 4 |
| Pre-processing | 4 |
| Strategies to use the partial label | 6 |
| Strategies to use the unlabeled images. | None |
| Strategies to improve model inference | 8 |
| Post-processing | 8 |
| Dataset and evaluation metric section is presented | 9 |
| Environment setting table is provided | 9 |
| Training protocol table is provided | 10 |
| Ablation study | 9 - 11 |
| Efficiency evaluation results are provided | 12 |
| Visualized segmentation example is provided | 13 |
| Limitation and future work are presented | Yes |
| Reference format is consistent. | Yes |