# OpenReview forum: "Two-stage training for abdominal pan-cancer segmentation in weak label"
_MICCAI.org/2023/FLARE — Submitted to FLARE 2023_

### Official Review · Reviewer_dR8Q · 2023-09-19
**straightforward and clear**

**Rating:** 6
**Confidence:** 4

**Review:**

Pros:
1.The article has a complete structure.
2.The method and process are clearly explained.

Cons:
1.Part abstract is too lengthy and can be further compressed.
2.Not all of the weak label described in the methods section of your article contain tumor label. After filling in, the weak label will become a label containing 13 types of organs or 14 types of organs and tumors. This cannot be used as a fully supervised data for Net training.

---

> ### Author Response · Authors · 2023-11-07
>
> Thanks for your careful reviews. I will make explanation and modify our manuscript according to your reviews.
>
> 1, In order to let readers simply grasp the core of the article, we have made a necessary and detailed description in the abstract. Regarding the long opinions on the summary, we have made the following modifications. Some of the content are removed from the background introduction and make some long sentences more concise.
>
> 2, I will explain from two points that the problem of tumor labels does not exist in all supplement weak label.
> First of all, we can't confirm whether each case contains a tumor, and the supplementation of the tumor area is unreliable to label. Based on this view, label containing 13 organs and all 14 classes both can be regarded as strong labels. At the same time, we did not use tumor labels for training in the pseudo-label generation network training stage. Therefore, the pseudo-label generation network  cannot generate tumor pseudo-labels and cannot be filled accordingly. Depending on your suggestion, we have further thought about the supplementary strategy of tumor labels and added them to the limit work.

---

### Official Review · Reviewer_jLwW · 2023-09-22
**Two-stage training for abdominal pan-cancer segmentation in weak label**

**Rating:** 7
**Confidence:** 5

**Review:**

The authors propose a two-stage training method to train an efficient segmentation model with weak labels. In the first stage, the models are trained using only strong labels, which are complete organ labels, employing the nnU-Net framework. In the second stage, a lightweight coarse-to-fine network is trained using the supplemented data, including both strong and pseudo labels.

---

> ### Author Response · Authors · 2023-11-03
>
> Thanks for your careful reviews and comments.

---

### Official Review · Reviewer_GSay · 2023-10-04
**This paper is good but lacks experiment setting details.**

**Rating:** 6
**Confidence:** 5

**Review:**

This paper is in general good shape, but lacks some critical details about experiments, including:

1. Dataset usage details, cross-validation, split info, etc.
2. Expeirment setting and training strategy details.
3. Evaluation details.

---

> ### Author Response · Authors · 2023-11-07
>
> Thanks for your careful reviews. I will make explanation and modify our manuscript according to your reviews.
>
> 1, Regarding the use of dataset, We have added more details in section 3.2. There is no  multiple cross-validation for the training of nn-Unet and EfficientSegNet. For the nnU-net, 1/5 of 222 cases was randomly selected as the validation set. For the EfficientSegNet, 100 cases in 2200 cases were randomly selected as the validation set.
>
> 2, Expeirment setting has been described in tabel 2 and tabel3. Depending on your suggestion，we supply futher details in section 3.2. About training strategy, we do more describ on loss strategy, optimizer setting and so on.
>
> 3, About evaluaton, we have further explained the possible reasons for the results of the data. In the qualitative analysis stage, the generation of bad case combined with our method has been further analyzed. You can see the relevant supplements in section 4.2.

---

### Official Review · Reviewer_cFAp · 2023-10-17
**Reviews**

**Rating:** 7
**Confidence:** 4

**Review:**

This paper reviews a study focused on a novel two-stage training method for medical image segmentation using weak labels. The proposed method leverages pseudo-labels to enhance training on incomplete datasets, and utilizes a lightweight coarse-to-fine network to optimize computational efficiency without compromising performance. Performance validation, conducted on the FLARE23 challenge dataset, indicates promising results with reduced computation time and high metric scores.

The only concern is how to ensure the reliability of weak labels for tumors.

---

> ### Author Response · Authors · 2023-11-03
>
> Thanks for your careful reviews. I will make explanation and modify our manuscript according to your reviews.
> For the reliability of weak labels for tumors, do you mean the weak label carried by the dataset itself or the weak label supplied with the pseudo-label? If it is the weak label of the dataset itself, the tumor label is all manual, we believe this part is reliable. For the supplement weak label, we did not supplement the tumor label. That is to say, tumor labels still have only reliable artificial labels.

---

### Decision · Program_Chairs · 2023-10-24

Accept